# Potentially Virulent Multi-Drug Resistant *Escherichia fergusonii* Isolated from Inanimate Surface in a Medical University: *Omphisa fuscidentalis* as an Alternative for Bacterial Virulence Determination

**DOI:** 10.3390/diagnostics13020279

**Published:** 2023-01-11

**Authors:** Noor Andryan Ilsan, Melda Yunita, Nurul Kusuma Dewi, Lalu Muhammad Irham, Siti Nurfajriah, Maulin Inggraini

**Affiliations:** 1Department of Medical Laboratory Technology, STIKes Mitra Keluarga, Bekasi 17113, Indonesia; 2Department of Medical Education, Faculty of Medicine, Universitas Pattimura, Poka 97233, Indonesia; 3Department of Biology, Faculty of Teacher Training and Education, Universitas PGRI Madiun, Madiun 63118, Indonesia; 4Faculty of Pharmacy, Universitas Ahmad Dahlan, Yogyakarta 55164, Indonesia; 5Department of Biology, Faculty of Mathematics and Natural Science, University of Bengkulu, Bengkulu 38122, Indonesia

**Keywords:** multi-drug resistant, virulent bacteria, Escherichia fergusonii, Omphisa fuscidentalis, Galleria mellonella

## Abstract

Multi-drug resistant (MDR) bacteria are becoming a worldwide problem due to limited options for treatment. Moreover, patients infected by MDR with highly virulent accessories are worsening the symptoms, even to the point of causing death. In this study, we isolated bacteria from 14 inanimate surfaces that could potentially be reservoirs for the spread of bacterial infections in the medical university. Blood agar media was used for bacterial isolation. The bacterial colony that showed hemolytic activities on each surface was tested for antimicrobial susceptibility against eight different antibiotics. We found that MDR bacterium, namely TB1, which was isolated from a toilet bowl, was non-susceptible to ampicillin, imipenem, chloramphenicol, amoxicillin-clavulanic acid, gentamicin, and tetracycline. Another MDR bacterium isolated from the mobile phone screen of security officers, namely HSO, was resistant to chloramphenicol, gentamicin, tetracycline, and cefixime. An in vivo virulence test of bacterial isolates used *Omphisa fuscidentalis* larvae as an alternative to *Galleria mellonella* larvae for the infection model. A virulence test of TB1 in *O. fuscidentalis* larvae revealed 20% survival in the bacterial density of 10^4^ and 10^5^ CFU/larvae; and 0% survival in the bacterial density of 10^6^ CFU/larvae at 24 h after injection. Bacterial identification was performed for TB1 as a potential virulent isolate. Bacterial identification using partial 16s rRNA gene showed that TB1 exhibited 99.84% identity to *Escherichia fergusonii* 2611. This study concludes that TB1 is a potentially virulent MDR *E. fergusonii* isolated from toilet bowls at a medical university.

## 1. Introduction

Numerous microorganisms present in daily life may contaminate important things and be able to cause infections in humans. Most people think that unclean hospital environments are the only places where there is a high risk for bacterial contamination. Nosocomial infections are infections that develop in hospitals, healthcare facilities, or other areas where pathogenic bacteria thrive. Because of the bacterial presence in those places, nosocomial infections can be spread to staff members, guests, or those with weak immune systems. Pathogenic bacteria can be in the air, water, soil, food, and nearby objects, such as mobile phones, computer equipment, and motorcycle handlebars [1].

Many bacteria, such as Gram-positive cocci (*Staphylococcus* spp., *Micrococcus* spp.), spore-forming rods (*Bacillus* spp.), or Gram-negative bacteria, can be transmitted through gadgets such as mobile phones or computer devices and cause nosocomial infections. Moreover, another source can serve as cellular reservoirs for pathogenic microorganisms, e.g., motorbike handlebars, which have a significant risk of causing nosocomial infections. Microbial transmission can also be spread from frequently used items such as mobile phones, computers, and motorbike handlebars that are not routinely disinfected [2]. Food that is consumed may become contaminated as a result of microbial transmission. In general, bacterial dissemination may occur when people come into direct or indirect contact with the bacterial source. In addition, droplet transmission may occur through the respiratory tract with large droplets [3].

Two important factors for determining the effect of pathogenic bacteria on the host are antimicrobial susceptibility and virulence. Antimicrobial susceptibility test results will guide the medical doctor in choosing a suitable treatment. Recently, we found bacteria that are resistant to many antimicrobial classes, termed as multi-drug resistant, extensively drug resistant, pan drug resistant [4]. These bacteria are difficult to treat due to the limited options for treatment. Nowadays, nosocomial bacterial infections can be resistant to many antibiotics. In the Middle East, nosocomial bacterial infections were found resistant to penicillin, cephalosporin, carbapenem, and floroquinolone, except for colistin [5]. A study in China showed that *E. coli* is the dominant nosocomial pathogen (859 isolates) with distinct antimicrobial profiles depending on its species [6].

Bacterial virulence is an important factor that will affect the severity of the disease caused by bacteria. Virulence factors in bacteria can include philus, fimbriae, iron-chelating molecule, and capsules [7]. Determining bacterial virulence can be performed in vivo on animals, such as mice, zebrafish, the nematode *C. elegans,* or the invertebrate *Galleria mellonella* [8]. *G. mellonella* larvae have a similarity to human defenses, especially innate immunity. Many studies have so far revealed the reliability of *G. mellonella* usage in bacterial virulence tests [9,10]. *G. mellonella* and *Omphisa fuscidentalis* are in the same taxonomic order, i.e., lepidoptera. Similar to *G. mellonella*, *O. fuscidentalis* has cellular and humoral immune responses so that we can evaluate the virulence of certain pathogens in their bodies. In this study, we used many bacterial sources from inanimate surfaces at a medical university as samples for bacterial isolation. One of the most frequent colonies on blood agar in each sample was chosen for an antimicrobial susceptibility test. We select one of these multi-drug resistant bacteria to perform a virulence test on *Omphisa fuscidentalis* larvae.

## 2. Materials and Methods

### 2.1. Bacterial Isolation and Antimicrobial Susceptibility Testing

The schematic method of this study is presented in Figure 1. Potentially nosocomial surface sources in one of the medical universities included the security officer’s and parking officer’s mobile phones, the toilet bowl, the motorcycle handlebar, the sauce bottle in the campus canteen, the storefront campus canteen, the parking office’s keyboard, the parking office’s computer mouse, and the parking gate button. Blood agar was used for bacterial isolation. This medical university is near its affiliated hospital. Furthermore, the lecturers and students have numerous contacts with the hospital for educational purposes. A sterile cotton bud was used for swabbing the surface, and then it was directly swabbed onto the blood agar (5%) surface [11]. The most dominant colony grown in each sample was selected for the antimicrobial susceptibility test. This test was performed following CLSI 2018. The selected bacteria were streaked onto Mueller-Hinton agar (MHA) with overnight incubation as a starter culture. The colony grown on Mueller-Hinton agar was streaked with a sterile cotton bud and resuspended in 5 mL of 0.8% NaCl in the glass tube. Bacterial suspension was adjusted to 0.5 Mac Farland using a Genesys 10S Uv-Vis spectrophotometer (Thermo Fisher Scientific, Waltham, MA, USA) at 600 nm wavelength. Adjusted bacterial suspension was then streaked on all of the surface of new Mueller-Hinton agar, then eight different antibiotics disc were placed onto MHA containing bacteria. Antibiotics that were used including ampicillin, imipenem, chloramphenicol, amoxycillin clavulanic acid, gentamicin, tetracycline, rifampicin, and cefixime. Bacteria and an antibiotic disc on MHA were incubated at 37 °C for 18 h. The diameter of the inhibition zone was calculated in mm. The resistant interpretation was compared with CLSI 2018.

### 2.2. Molecular Identification of Isolate KS-1 and Phylogenetic Tree Construction

The genomic DNA of isolate KS-1 was extracted by boiling at 98 °C for 15 min in a thermal cycler. The genomic DNA of isolate KS-1 was further used as a template for Polymerase Chain Reaction (PCR) amplification of partial 16S rRNA sequence using 1387r primer (5′ GGGCGGWGTGTACAAGGC 3′) and 63f primer (5′ CAGGCCTAACACATGCAAGTC 3′) [12]. The amplicon product of PCR was 1300 bp. A 50 µL total reaction of PCR mixture was prepared with the composition as follows: 25 µL GoTaq Green Master Mix (Promega, Madison, WI, USA), 5 µL primer 1387r (10 pmol), 5 µL primer 63f (10 pmol), 4 µL colony-boiled genomic DNA of KS-1, and 11 µL nuclease-free water. The PCR was conditioned in 30 cycles, with pre denaturation at 94 °C for 5 min, denaturation at 94 °C for 30 s, annealing at 55 °C for 4 s, and elongation at 72 °C for 10 min. A total of 1.5% agarose gel was used for separating the DNA product. The separated DNA band was visualized under UV a transilluminator. Subsequently, PCR product was sequenced using a Sanger method in 1^st^ Base Genetika Science, Indonesia. All sequences from the forward and reverse primers were trimmed and assembled using MEGA11 The assembled sequence was aligned using the BLASTN method on the National Center for Biotechnology Information (NCBI) website against the closest reference. A phylogenetic tree was constructed using Mega X version 11 using the maximum likelihood statistical method. The best analysis model for all partial 16S rRNA sequences was determined using Find Best Model menu, which found that Tamura-Nei (TN93) was a suitable model.

### 2.3. Culture Preparation of Escherichia fergusonii for Omphisa fuscidentalis Injection

Culture preparation followed Ilsan et al. [10]. *E. fergusonii* was cultured on BHI agar. Grown colonies of *E. fergusonii* were subcultured into 2 mL of sterile BHI broth, and they were then incubated using a shaker incubator with a speed of 100 RPM of overnight. A total of 1 mL of bacterial liquid culture was centrifuged at 8000 RPM for 5 min at room temperature. The supernatant was disposed, and the pellet was re-suspended with 500 µL sterile-phosphate buffer saline (PBS). Bacterial suspension was measured for its absorbance at OD600 using a Genesys 10S Uv-Vis spectrophotometer (Thermo Fisher Scientific, Waltham, MA, USA). The bacterial suspension was adjusted with PBS to an OD of 1, with a bacterial density of approximately 10^9^ CFU/mL. The injection dose of bacterial suspension used was 10^7^, 10^6^, 10^5^, 10^4^ CFU/larva.

### 2.4. Escherichia fergusonii Injection in Omphisa fuscidentalis Larvae as an Infection Model

The bacterial injection method refers to Ilsan [10] who previously used *Galleria mellonella* larvae infected by *Acinetobacter baumannii.* In this study, *O. fuscidentalis* larvae were reared in homegrown. Late phase of *O. fuscidentalis* larvae with a size of 300–400 g was used in this study. Ten replicates of larvae were injected for each group, including a PBS-injected control. Prior to injection, those larvae were placed on alcohol-soaked tissues. Furthermore, a total of 10 µL bacterial suspension was injected into the last left proleg of the larvae. All of the injections were performed in the JSCB-900SB Biosafety Cabinet (JSR, Gongju, Republic of Korea). After injection, those larvae were incubated at 37 °C for 8 h and 24 h. Survival percentage were evaluated at 8 and 24 h after incubation [9]. The Kaplan-Meier survival curve was constructed using GraphPad Prism 5 (GraphPad, San Diego, CA, USA). The Log-rank statistical analysis of the Kaplan-Meier survival curve was performed using GraphPad Prism 5. The statistical significance of the melanization score was determined using one-way ANOVA in GraphPad Prism 5.

## 3. Results

This study showed that all of the surfaces had culturable bacteria on blood agar isolation. Most dominant colonies grown with hemolytic accessories were continued for antimicrobial susceptibility test against eight antibiotics using disk diffusion method. All bacteria were non-susceptible to cefixime with 50% prevalence (7/14), followed by chloramphenicol, ampicillin, imipenem, amoxicillin clavulanic acid, gentamicin, and tetracycline with 21.4% (3/14) (Table 1).

Two bacterial isolates isolated from a toilet bowl and storefront canteen were resistant to imipenem, which is considered a last-line antibiotic. According to Magiorakos [4], isolate TB1 from toilet bowl 1 was categorized as multi-drug resistant (MDR), which is non-susceptible to six antibiotic classes (Figure 2). While bacteria isolated from the mobile phone of the security officer was non-susceptible to four antibiotic classes. Isolate TB1 was then studied further for molecular identification and virulence test in *Omphisa fuscidentalis* larvae as a bacterial infection model.

Molecular identification of isolate TB1 using partial 16s rRNA showed that TB1 is closely related to *Escherichia fergusonii* strain 2611 with 99% query cover and 99.84% identity in 1438 bp length (Table 2).

The MDR *E. fergusonii* TB1 was then tested for bacterial virulence in *Omphisa fuscidentalis* larvae. A bacterial suspension dosage of 10^3^–10^7^ CFU/larvae was used for the virulence test. Survival and melanization observation were conducted in 4 h and 24 h after injection (Figure 3). In 4 h after injection of TB1, 10^6^ CFU and 10^7^ CFU/larvae had 70% and 60% larvae survival, respectively. Meanwhile, a bacterial suspension dosage of 10^3^–10^5^ CFU showed no larvae dead. In the 24 h after injection, 10^6^ CFU and 10^7^ CFU/larvae had 0% survival, which means all of 10 larvae were dead. While, 10^5^ CFU and 10^6^ CFU/larvae had 20% survival. Moreover, 10^3^ CFU/larvae had 30% survival proportion (Figure 4). Statistical analysis of Kaplan-Meier survival curve using Logrank test for trend showed all these five dosage survival curves are in a linear trend with P 0.0061. This indicates that higher dosages of bacterial suspension cause higher mortality in the larvae.

## 4. Discussion

The inanimate surface, especially in a hospital or medical university, can be a reservoir for bacterial contamination. Pathogenic bacteria can be found on medical charts, bed rails, the surface of the mobile phone screen, the CPU keyboard and mouse, and water sinks [13,14,15]. Highly resistant pathogenic bacteria with virulence accessories have been found on inanimate surfaces in medical settings, including *Acinetobacter baumannii*, *Pseudomonas aeruginosa*, *Klebsiella pneumoniae*, *Escherichia coli*, methicillin-resistant *Staphylococcus aureus*, and vancomycin-resistant *Enterococci* [16]. Factors that affect the transfer and survival of bacterial attachment on the surface including species, surface type, surface humidity, number of inoculums, hand hygiene of the user, ward design, number of infected patients, and proper antibiotic usage [17,18]. A meta-analysis from 18 studies in Ethiopia showed 70% of inanimate surfaces and equipment were contaminated by bacteria [19].

In this study, we used several surfaces as potential sources of bacterial contamination. The mobile phones of healthcare workers in the university teaching hospital in Zambia showed a 79% prevalence of bacterial contamination. The predominant isolates were coagulase-negative *Staphylococci*, *S. aureus, Bacillus* spp., and *E. coli.* The majority of those isolates were susceptible to cotrimoxazole, gentamicin, and tetracycline. In this study, isolate HSO from mobile phone of security officer was resistant to chloramphenicol, gentamicin, tetracycline, and cefixime. Resistance to these antimicrobials was categorized as a MDR bacteria. MDR bacteria, according to Magiorakos et al. [4], are bacteria that are resistant to more than three antimicrobial classes

We found hemolytic bacteria on the motorcycle handlebars of staff and the outer surface of the canteen’s bottle sauce. These bacteria were susceptible to eight antimicrobial tests. Bacterial isolates were also found on parking keyboard and mouse of this medical university. Only one isolate was resistant to cefixime, while the rest of antimicrobials tested were susceptible. Nazeri et al. [20] reported that 76% of computer keyboard and electronic equipment were contaminated by bacteria. This study was conducted in an ICU hospital in Iran.

The outbreaks of bacterial infection have been reported to take place in toilets. Recently, the outbreaks happened in the toilet’s hospital involves MRSA and *Legionella pneumophila* [21]. *Salmonella enteridis* has been reported to develop biofilm in the toilet bowl of four patients recovering from salmonellosis [22]. Biofilm was found under the rim of the toilet bowl. In this study, we isolated MDR bacterium that was obtained from toilet bowl and identified as *Escherichia fergusonii* based on their partial 16s rRNA sequence. As our knowledge, this study is the first report of MDR *E. fergusonii* from inanimate surface.

*E. fergusonii* is closely related to *E. coli* with 60% similarity. *E. fergusonii* was announced as a new species in 1985 [23]. *E. fergusonii* is an opportunistic pathogen associated with several sites in humans and animals. In humans, this bacterium causes bacteremia, UTIs, and abdominal wounds. While in animals, it leads to septicemia and diarrhea [24,25,26,27]. *E. fergusonii* attracts worldwide attention due to its resistance to several antibiotics. The first report of ESBL production by *E. fergusonii* was reported in 2010 [28]. Furthermore, carbapenem-resistant *E. fergusonii* strains containing beta-lactamase genes have been reported for the first time in 2019 from clinical specimens [29]. TB1 MDR *E. fergusonii* also had hemolytic activity on blood agar. This is likely evidence that *E. fergusonii* may cause hemolytic uremic syndrome [30].

A virulence test is an important step for determining the severity of the symptoms experienced by the host. *Galleria mellonella* is the invertebrate well-established bacterial infection model so far with more than 2200 publications [31]. We used an alternative model namely *Omphisa fuscidentalis* instead of *Galleria mellonella* as they are lepidopteran. They have many similarities in term of immune defense systems. *Galleria mellonella* is the earliest species used for study the immunity in insect [32]. However, some studies explore the immune response of the specific order, lepodoptera. *Galleria mellonella* and *Omphisa fuscidentalis* are lepidopteran group. Pathogenic bacteria produce proteinases enzyme to occupy lepidopteran protein in hemolymph as a source for nutrients and undergoes metabolism. Proteinases produced by pathogenic bacteria has function to degrade the antimicrobial peptides that is part of lepidopteran immunity [32]. The skin, or epithelial cells of lepidopteran larvae are the frontline barrier between hemolymph and the environment. The open wound on epithelia causes a hemolymph clot in this region [33]. Hemokine and chemotactic peptide, which are released by damaged ephitelial cells, may represent signaling or adhesion molecules that trigger aggregation of hemocytes [34]. Plasmatocytes, part of hemocyte cell, are ruptured and released into hemolymph throughout the process, which causes an extracellular matrix to form a soft clot seals the wound [35]. Following activation of the transglutaminase/pro-phenoloxidase (PPO) cascade, the clot is cross-linked and melanization occur become a hard clot [36]. Overall, clot production is a crucial part of the insect immune system’s defense. A clot confines bacteria at the site of wound, preventing them from traveling to the haemocoel and infection surrounding tissues in addition to promoting wound healing and minimizing hemolymph loss. Additionally, the phenoloxidase (PO) system’s activation reinforces the killing and removal of the entrapped microbes [37]. Moreover, in response to microbial invasion or integument injury in lepidopteran, hemocytes actively participate in the synthesis of a variety of antimicrobial peptides (AMPs) and proteins that are discharged into the hemolymph. The primary classes of AMPs, which are further divided according on the secondary structures and sequence composition, are expressed by hemocytes. These classes include cecropins, linear amphipathic alpha-helical AMPs, defensins, and AMPs with high proline and glycine content. In response to microbial stimulation, hemocytes also produce a variety of antimicrobial proteins, such as lysozymes [38], transferrins [39], and a variety of soluble microbial pattern recognition receptors, such as C-type lectins, peptidoglycan recognition proteins (PGRPs), -1,3-glucan recognition proteins (GRPs), and galectins (GALEs) [40,41].

In addition, Lepidopterans lack an adaptive immune system like other insects, but their innate immune system is very similar to that of mammals. Hemocytes, immune cells related to animal neutrophils, are important players in the cellular response, which also involves a humoral response with soluble effector molecules. Hemocytes are primarily located in hemolymph, which is the equivalent of mammalian blood, although they are also subcuticular, dispersed throughout the fatty body, and near the digestive system. Throughout life, hemocyte concentration changes, and stress brought on by microbes also has an impact [42]. They can also release extracellular nucleic acid traps, which are involved in the sequestration of microorganisms and the stimulation of coagulation, like mammal neutrophils [43].

For *E. fergusonii* TB1 virulence testing, bacterial suspension of 10^3–^10^7^ CFU was used. Survival and melanization score were evaluated in 4 h and 24 h after injection. In the 24 h after injection, 10^6^ and 10^7^ had 0% survival, while 10^5^ had 20% survival. Statistical analysis of Kaplan-Meier survival curve using Logrank test for trend showed all these five dosage survival curves is in a linear trend. Higher dosage of bacterial suspension caused higher mortality of the larvae. So far, there is no study conducting virulence tests using *O. fuscidentalis.* Virulence test of Enteropathogenic *E. coli* (EPEC) using *G. mellonella,* started at 4 × 10^4^ CFU dosage, which led to 0% survival [44].

The cellular immune system and the humoral immune system are the two main immune systems found in insects. Phagocyte cells, also known as hemocytes, found in hemolymph mediate the cellular immune system in larvae. In addition to having phagocytic function, hemocytes in larvae also enclose and clot the foreign invaders. In addition to serving as complement-like substances, antimicrobial peptides, and melanin, soluble compounds that operate as mediators of the humoral immune system also trap microorganisms [9]. *O. fuscidentalis* infected by *E. fergusonii* TB1 likely produced melanization or blackness by depositing and synthesizing melanin to capture or encapsulate the bacteria along with hemolymph opsonization and coagulation. This phenomenon was strongly tied to the growth of abscesses in mammalian bacterial infections. In hemocytes, phenoloxidase produces melanin. Reactive oxygen species produced by phenoloxidase have also been linked to bacterial harm.

## 5. Conclusions

In conclusion, multi-drug resistant *E. fergusonii* TB1 has been found in a toilet bowl at a medical university. Moreover, *E. fergusonii* TB1 showed an MDR phenotype, including resistance to carbapenem, which is considered a last-line antibiotic. For virulence testing, we consider *O. fuscidentalis* as an alternative for *G. mellonella* larvae as a well-established invertebrate infection model. TB1 bacterial suspension of 10^5^ CFU led to 0% survival in the 24 h after injection. More research into O. fuscidentalis as an infection model is required.

## Figures and Tables

**Figure 1 diagnostics-13-00279-f001:**
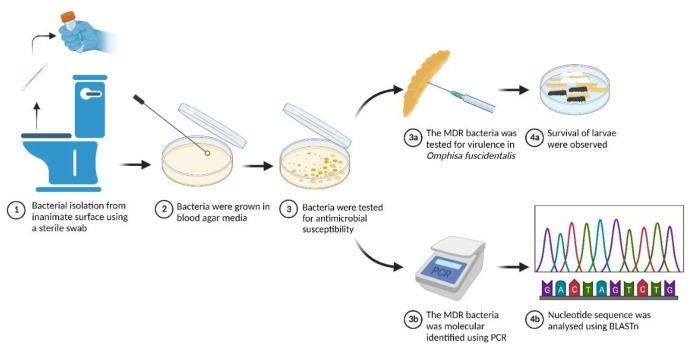
The schematic method of this study is illustrated in this figure. Bacteria were isolated from inanimate surfaces of various sources at a medical university on blood agar. The grown bacteria were then tested for antimicrobial susceptibility. The MDR bacteria, TB1 isolate, was then further tested for molecular identification and virulence determination. This figure was created with BioRender.com under agreement number “JH24RZTNDZ“.

**Figure 2 diagnostics-13-00279-f002:**
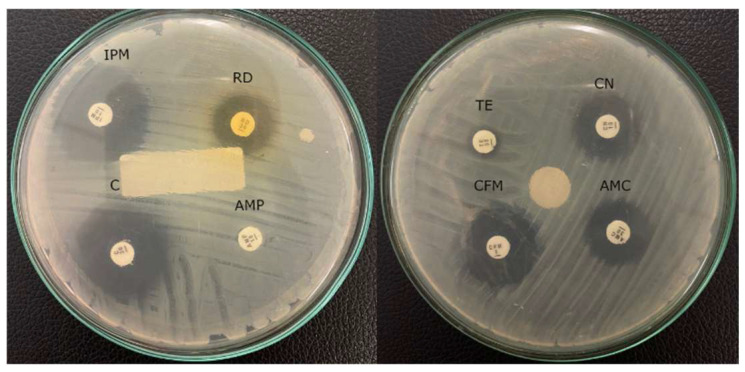
Antimicrobial susceptibility test result of TB1 isolate against eight different antibiotics using disk diffusion method.

**Figure 3 diagnostics-13-00279-f003:**
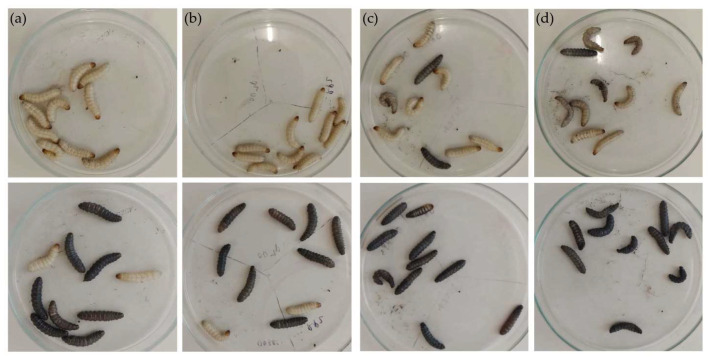
*Omphisa fuscidentalis* larvae after being injected by *E. fergusonii* TB1 suspension in 10^4^–10^7^ CFU/larvae. The upper figure is 4 h after injection, and the lower figure is 24 h after injection. (**a**) 10^4^, (**b**) 10^5^, (**c**) 10^6^, (**d**) 10^7^ CFU/larvae.

**Figure 4 diagnostics-13-00279-f004:**
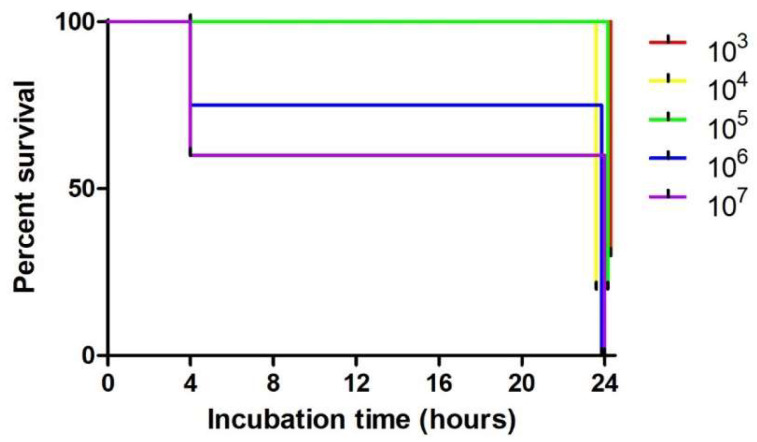
Kaplan-Meier survival curve of *Omphisa fuscidentalis* that was injected by *E. fergusonii* TB1 in 4 and 24 h after incubation. Dosage unit is CFU/larvae.

**Table 1 diagnostics-13-00279-t001:** Antimicrobial susceptibility testing of bacterial isolates against eight different antibiotics using the disk diffusion method.

Surface of Samples (Isolate Code)	Diameter of Inhibition Zone (mm)							
	AMP 10	IPM 10	C 30	AMC 30	CN 10	TE 30	RD 30	CFM 5
Mobile phone of security officer (HSO)	20 (S)	45 (S)	0 (R)	29 (S)	10 (R)	10 (R)	30	0 (R)
Mobile phone of parking officer (HPO)	30 (S)	30 (S)	44 (S)	42 (S)	44 (S)	30 (S)	30	40 (S)
Toilet bowl 1 (TB1)	0 (R)	17 (R)	16 (I)	16 (I)	12 (R)	8 (R)	15	20 (S)
Toilet bowl 2 (TB2)	16 (I)	32 (S)	26 (S)	12 (R)	27 (S)	33 (S)	35	0 (R)
Motorcycle handlebar 1 (MH1)	36 (S)	50 (S)	30 (S)	38 (S)	26 (S)	30 (S)	44	8 (S)
Motorcycle handlebar 2 (MH2)	35 (S)	40 (S)	24 (S)	40 (S)	27 (S)	30 (S)	40	8 (S)
Sauce bottle in canteen 1 (SBIC1)	25 (S)	52 (S)	34 (S)	34 (S)	25 (S)	30 (S)	40	26 (S)
Sauce bottle in canteen 2 (SBIC2)	27 (S)	50 (S)	30 (S)	32 (S)	25 (S)	33 (S)	40	24 (S)
Storefront canteen 1 (SC1)	23 (S)	56 (S)	23 (S)	36 (S)	26 (S)	31 (S)	58	26 (S)
Storefront canteen 2 (SC2)	23 (S)	0 (R)	32 (S)	34 (S)	30 (S)	31 (S)	0	17 (I)
Parking keyboard 1 (PK1)	23 (S)	50 (S)	25 (S)	34 (S)	28 (S)	30 (S)	44	20 (S)
Parking keyboard 2 (PK2)	22 (S)	42 (S)	24 (S)	32 (S)	28 (S)	34 (S)	30	0 (R)
Parking computer mouse (PCM)	15 (S)	32 (S)	30 (S)	26 (S)	22 (S)	23 (S)	0	0 (R)
Parking ticket button (PTB)	18 (S)	47 (S)	26 (S)	23 (S)	23 (S)	20 (S)	28	0 (R)

AMP: Ampicillin, IPM: Imipenem, C: Chloramphenicol, AMC: Amoxicillin-clavulanic acid, CN: Gentamicin, TE: Tetracycline, RD: Rifampicin, CFM: Cefixime. Resistant interpretation followed CLSI 2018 with *Enterobacteriaceae* criteria. Interpretation of Rifampicin is not mentioned in CLSI 2018.

**Table 2 diagnostics-13-00279-t002:** BLASTN result of partial 16s rRNA gene sequence of TB1. The description below is only of the three closest strains in NCBI.

Description	Max Score	Query Cover	E Value	Identity	Accession Length (bp)	Accession
*Escherichia fergusonii* strain 2611 16S ribosomal RNA gene	2361	99%	0.0	99.84%	1438	MT611634.1
*Escherichia fergusonii* strain 389 16S ribosomal RNA gene	2361	99%	0.0	99.84%	1384	MT573069.1
*Escherichia fergusonii* strain 346 16S ribosomal RNA gene	2361	99%	0.0	99.92%	1376	MT573049.1

## Data Availability

Not applicable.

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
