# Peer review of "Potentially Virulent Multi-Drug Resistant Escherichia fergusonii Isolated from Inanimate Surface in a Medical University: Omphisa fuscidentalis as an Alternative for Bacterial Virulence Determination"

_diagnostics, 2023, doi:10.3390/diagnostics13020279_

Round 1

Reviewer 1 Report

diagnostics-2092403 Potentially Virulent Multi-drug Resistant Escherichia fergusonii Isolated From Inanimate Surface in a Medical University: Omphisa fuscidentalis as an Alternative for Bacterial Virulence Determination

Authors described the identification of Escherichia fergusonii, potential multi-drug resistant bacteria, from inanimate surface and the potential use of Omphisa fuscidentalis as a research model. The topic of manuscript and idea of research is innovative. However, I have impression that the text is chaotic. First of all there is lots of spelling mistake, for example callular despite cellular. Please read carefully the whole text. Also I think that the extensive editing of english language and style is required. Next you described about the advantages of G. mellonella larvae as alternative research model, and then that you will use the another insect, Omphisa fuscidentalis, in your research. Please explain why you think that this model will be also good in infection research (I mean, write more about the cellular and humoral immune response of both insects, what are the similarities and differences between them). Moreover, argument that: "G. mellonella larvae is not be sold commercially" is weak, becouse it that kind of research you need larvae rearing in laboratory conditions, cultivating for few generation, to obtain similar results. Also I have problem with this sentence: "So that we used an alternative invertebrate model that has a close relationship..", but what kind of relationship you mean? In my opinion even if there is close genetic relationship, there is strong need to describe more about the immunological similarities between two species. Next, in method section you write about the hemocyte staining, however I did not see the result and discussion of this results.

Author Response

Thank you for your deep review of our manuscript Prof.

Our pleasure to have you as our reviewer

We revise your recommendation as idea below:

1. We realize that our grammar writing is not good and spelling mistake, hence we proofreaded our manuscript for english editing, the revision is marked in Track change menu in word file
2. As your recommendation, we deleted "G. mellonella larvae is not be sold commercially", and "So that we used an alternative invertebrate model that has a close relationship.." is not so strong idea, furthermore, we added more about the similarity of G. mellonella and O. fuscidentalis immunity as lepidopteran. Here is our additional text "A virulence test is an important step for determining the severity of the symptoms experienced by the host. Galleria mellonella is the invertebrate well-established bacterial infection model so far with more than 2200 publications [31]. We used an alternative model namely Omphisa fuscidentalis instead of Galleria mellonella as they are lepidopteran. They have many similarities in term of immune defense systems. Galleria mellonella is the earliest species used for study the immunity in insect [32]. However, some studies explore the immune response of the specific order, lepodoptera. Galleria mellonella and Omphisa fuscidentalis are lepidopteran group. Pathogenic bacteria produce proteinases enzyme to occupy lepidopteran protein in hemolymph as a source for nutrients and undergoes metabolism. Proteinases produced by pathogenic bacteria has function to degrade the antimicrobial peptides that is part of lepidopteran immunity [32]. The skin, or epithelial cells of lepidopteran larvae are the frontline barrier between hemolymph and the environment. The open wound on epithelia causes hemolymph clot in this region [33]. Hemokine and chemotactic peptide, which are released by damaged ephitelial cells, may represent signalling or adhesion molecules that trigger aggregation of hemocytes [34]. Granulocytes, part of hemocyte cell, are ruptured and released into hemolymph throughout the process, which causes an extracellular matrix to form a soft clot seals the wound [35]. Following activation of the transglutaminase/pro-phenoloxidase (PPO) cascade, the clot is cross-linked and melanization occur become a hard clot [36]. Overall, clot production is a crucial part of the insect immune system’s defense. A clot confines bacteria at the site of wound, preventing them from traveling to the haemocoel and infection surrounding tissues in addition to promoting wound healing and minimizing heamolymph loss. Additionally, the phenoloxidase (PO) system’s activation reinforces the killing and removal of the entrapped microbes [37]. Besides, In response to microbial invasion or integument injury in lepidopteran, haemocytes actively participate in the synthesis of a variety of antimicrobial peptides (AMPs) and proteins that are discharged into the haemolymph. The primary classes of AMPs, which are further divided according on the secondary structures and sequence composition, are expressed by hemoglobinocytes. These classes include cecropins, linear amphipathic alpha-helical AMPs, defensins, and AMPs with high proline and glycine content. In response to microbial stimulation, haemocytes also produce a variety of antimicrobial proteins, such as lysozymes [38], transferrins [39], and a variety of soluble microbial pattern recognition receptors, such as C-type lectins, peptidoglycan recognition proteins (PGRPs), -1,3-glucan recognition proteins (GRPs), and galectins (GALEs) [40, 41]."

Reviewer 2 Report

The article is very interesting to read however few comments are below for the authors to consider:

1. as there is a separate materials section, it is advisable to add all of the materials, chemicals, etc in that section

2. if possible add the primers backward and forward into a small table for clarity

3. would it be possible to add the MIC for known antibiotics for both sensitive and resistant strains

4.  please state clearly the interest of the microbe E. fergusonii 

5. would it be possible to add a short paragraph about the species phylogeny

6. was there any gel electrophoresis analysis during the time of conduction of the work to evaluate the PCR product

Author Response

Thank you for your deep review of our manuscript Prof.

Our pleasure to have you as our reviewer

We revise your recommendation as idea below:

1. We added all of the materials including brand, and the company's city, for example ". The colony grown on Mueller-Hinton agar was streaked with a sterile cotton bud and resuspended in 5 ml of 0.8% NaCl in the glass tube. Bacterial suspension was adjusted to 0.5 Mac Farland using a Genesys 10S Uv-Vis spectrophotometer (Thermo Fisher Scientific, Waltham, MA, USA) at 600 nm wavelength"

2. Since we only use 1 pair of primer, so in our opinion is enough to write down on the text instead in the table. If we use many primers, it will be better presented in table

3. Since we used disk diffusion method, for the result we only have diameter clear zone (inhibition zone) only instead of MIC. As we know, MIC data is only valuated if we use dilution broth method. However, in CLSI standard, both disk diffusion and dilution broth are can be used.

4 and 5. We added description of E. fergusonii as "E. fergusonii is closely related to E. coli with 60% similarity. E. fergusonii was announced as a new species in 1985 [23]. E. fergusonii is an opportunistic pathogen associated with several sites in humans and animals. In humans, this bacterium causes bacteremia, UTIs, and abdominal wounds. While in animals, it leads to septicemia and diarrhea [24-27]. E. fergusonii attracts worldwide attention due to its resistance to several antibiotics. The first report of ESBL production by E. fergusonii was reported in 2010 [28]. Furthermore, carbapenem-resistant E. fergusonii strains containing beta-lactamase genes have been reported for the first time in 2019 from clinical specimens [29]. TB1 MDR E. fergusonii also had hemolytic activity on blood agar. This is likely evidence that E. fergusonii may cause hemolytic uremic syndrome [30]."

6. Yes we have it. We upload the figure in the attachment. Our isolate is in point 1

Round 2

Reviewer 1 Report

I am afraid that presented manuscript still have some mistakes. Authors put an effort to correct previous mentioned language mistake. However, I still have some doubts about this manuscript. You put long fragment about the defense system of insect, but based on your information it is still not clear why insect are good research model for this kind of research (I mean there is lack of information about the similarities between insect and human defense system, that your results really can be compare with mammals one). Also you still based your research on thesis that both of insect are lepidoptera and have cellular and humoral immune defense. That is very poor argument, there are plenty of lepidopteran insect with similar immunological defense. Please put some information about the characteristic of Omphisa fuscidentalis hemocyte class, the description of phagocytosis, lyzozyme role, PO release, etc. Then compare it with data from Galleria mellonella and mammals, and then you can put a thesis that this insect might be a good alternative model of infection. Moreover, there are some methodological mistake in fragment that you put, for example you use "hemoglobinocytes", what is that, there is reference with that sentence, and also I could not find any information about this term.  Also I did not agree to the role of granulocytes in clot formation, in my opinion the plazmatocytes are more important role in that process. 

Author Response

Thank you Prof for your detail suggestions

  1. Hemoglobinocytes: you are true, after we read carefully, the true one is hemocytes. So we changed it. Guessing of the mistake in automatic correction
  2. Granulocytes: After we read some references, we agree with you, plasmatocytes plays more important role for clotting. So we changed it, also the reference. Thank you Prof
  3. Similarity between lepidopteran and mamalian immune system: We added

"In addition, Lepidopterans lack an adaptive immune system like other insects, but their innate immune system is very similar to that of mammals. Hemocytes, immune cells related to animal neutrophils, are important players in the cellular response, which also involves a humoral response with soluble effector molecules. Hemocytes are primarily located in hemolymph, which is the equivalent of mammalian blood, although they are also subcuticular, dispersed throughout the fatty body, and near the digestive system. Throughout life, hemocyte concentration changes, and stress brought on by microbes also has an impact [42]. They can also release extracellular nucleic acid traps, which are involved in the sequestration of microorganisms and the stimulation of coagulation, like mammal neutrophils [43]."

4. This is our personal statement. We aware this study have many weaknesses, including this is a new model for bacterial infection, instead of using G. mellonella. That is why se cannot find/difficult to find Omphisa fuscidentalis detail information for their immune system. So I make it general for lepidopteran. Hope another person or us, will study more detail about this model. Why we are optimistic because in another study of us, we used endophytic bacteria from mangrove leaves -it should be non-pathogenic trait-, and in 10^3 CFU O. fuscidentalis injection, we found most of the are survived, means it can be a potential model for study bacterial infection. Besides, more study need to be conducted. Thank you Prof

Round 3

Reviewer 1 Report

Authors responded to my concern.